# Fish provision in a changing environment: The buffering effect of regional trade networks

**Blanca González-Mon** [1]*, **Emilie Lindkvist**[1], **Örjan Bodin**[1], **José Alberto Zepeda-Domínguez**[2], **Maja Schlüter**[1]

1 Stockholm Resilience Centre, Stockholm University, Stockholm, Sweden, 2 Facultad de Ciencias Marinas, Universidad Autónoma de Baja California, Ensenada, México

* blanca.gonzalez@su.se

**Data Availability Statement:** The Small-Trade model and all input/output files are available from the CoMSES computational model library (https://doi.org/10.25937/n9z9-8n29).Please note that the

## Abstract

Local and regional trade networks in small-scale fisheries are important for food security and livelihoods across the world. Such networks consist of both economic flows and social relationships, which connect different production regions to different types of fish demand. The structure of such trade networks, and the actions that take place within them (e.g., people fishing, buying, selling), can influence the capacity of small-scale fisheries to provide sufficient fish in a changing social and ecological context. In this study, we aim to understand the importance of networks between different types of traders that access spatially-distinct fish stocks for the availability and variability of fish provision. We deployed a mixed-methods approach, combining agent-based modelling, network analysis and qualitative data from a small-scale fishery in Baja California Sur, Mexico. The empirical data allowed us to investigate the trade processes that occur within trade networks; and the generation of distinct, empirically-informed network structures. Formalized in an agent-based model, these network structures enable analysis of how different trade networks affect the dynamics of fish provision and the exploitation level of fish stocks. Model results reveal how trade strategies based on social relationships and species diversification can lead to spillover effects between fish species and fishing regions. We found that the proportion of different trader types and their spatial connectivity have the potential to increase fish provision. However, they can also increase overexploitation depending on the specific connectivity patterns and trader types. Moreover, increasing connectivity generally leads to positive outcomes for some individual traders, but this does not necessarily imply better outcomes at the system level. Overall, our model provides an empirically-grounded, stylized representation of a fisheries trading system, and reveals important trade-offs that should be considered when evaluating the potential effect of future changes in regional trade networks.

## 1. Introduction

Fish is one of the most internationally traded foods that contributes significantly to global food security [1]. At the same time, local trade is critical to guarantee the supply of nutritious food to diverse local consumers, and especially to low-income consumers [2,3]. Maintaining a stable

DOI links to a landing page with the general information of the model, and in that page there is green rectangle stating "Download Version 1.1.0" to the right of the screen. Clicking that rectangle will download a folder containing subfolders with the model code in Netlogo format, the data needed as input to run the model and an exemplar of the results and R code used to produce the paper figures, amongst other things. Those files will enable anyone to run the model, but note that you may need to save the Netlogo file and the data files in the same folder to run all scenarios successfully, as it is required by the Netlogo software by default. Interview data cannot be shared publicly because of confidentiality agreements. Anonymized data are available from the authors for researchers who meet the criteria for access to confidential data. Requests may be directed to src-ethics-review@su.se and to the corresponding author.

**Funding:** BG, EL and MS received funding from the European Research Council (ERC) under the European Union's Horizon 2020 research and innovation programme, grant agreement No 682472 — MUSES. EL and MS were funded by US National Science Foundation Coupled Natural and Human Systems Program (www.nsf.gov), DEB-1632648. EL was also funded by the Swedish Research Council (Vetenskapsrådet; www.vr.se), Dnr 2018-05862. ÖB received funding from the Swedish Research Council Formas (Dnr 2020-01551). The funders had no role in study design, data collection and analysis, decision to publish, or preparation of the manuscript.

**Competing interests:** The authors have declared that no competing interests exist.

fish supply is, however, challenging because of complex social, institutional, and ecological dynamics that make fisheries highly variable and uncertain. This includes seasonal variability that affects local supply chains and consumption [2], and longer-term environmental changes [4]. In the long term, guaranteeing food supply requires sustaining fish populations [2], given the many losses associated with overfishing [5,6]. Thus, it is essential to understand how to ensure fish provision in a changing social-ecological environment. In this context, trade and trade networks are key. Value chain and supply chain approaches have been extensively used to study trade relationships in fisheries over recent years [7,8], but how such regional trade structures deal with changes and influence the dynamics of fish availability has been rarely studied. In this manuscript, we investigate if and how regional (sub-national) patterns of trade among different types of small-scale fishery traders affect the overall capacity and variability of fish provision in the context of spatially heterogeneous, multi-species fisheries with changing fish availability.

Small-scale fisheries (SSF) are characterized by high diversity in terms of the species targeted, actors, institutions, and the decentralization of landing sites [9]. Fishing localities are often heterogeneous, which can be linked to ecological, biophysical or institutional factors [10]. The fish caught in diverse production regions are often directed towards central points of commercialization (e.g., cities), through networks of traders that consist of the regional (i.e., subnational or national) exchange of fish [11,12]. These networks are spatially determined creating networks of socio-spatial relationships through which different places interact [13,14]. They connect the diverse and decentralized producers with different market demands, from local consumption to global markets, which contributes to fish provision across scales [3,15,16]. This multi-market context often characterizes SSF, where different types of demand target specific species (e.g., [17]).

Trade relationships in SSF commonly take place in local and regional networks of traders [15,18–20], where trade is embedded in networks of social relationships [21]. Thus trade relationships in SSF are often interlinked with other social processes and relationships such as kinship, assistance, or informal institutions [20,22]. There are numerous examples describing stable relationships between fishers (producers) and traders as embedded in institutional arrangements often named patron-client relationships, which involve loans, loyalty or commitment, reciprocity, and/or the exchange of different goods and services (e.g., [22–25]). The same type of arrangements have been found between traders in regional supply chains [8,20]. In addition, the trade and social relations within trade networks can influence traders' decisions, which can be partly driven by the actions of their network peers. For instance, trade relations can influence which fish species are targeted, and how they are targeted, in response to market demands and social relationships between and amongst fishers and traders [17,24,26]. Trade relationships can also create spillover effects between different places. For example, they can channel market demands across space and/or trigger the mobility of traders, leading to patterns of sequential resource exploitation at a global scale, or mask signals of fisheries decline [24,27–29]. Thus, there is a need to better understand how regional trade networks associated with diverse geographies and species can influence these dynamics and impact fish provision in SSF.

In this context, it is essential to recognize the existence of diverse types of traders that have different roles and/or functions in these trade arrangements and networks [15,30–32]. These functions can be mediated by traders' positions in the regional trade networks [19,33]. The interplay between roles and positions affect traders' vulnerabilities and their supply of fish, potentially influencing their capacity to deal with changes and short-term supply fluctuations [15,34]. Studies on fisheries trade have often focused on understanding vertical interactions in the value chain (i.e., all stages from producers to consumers) rather than on trade relationships

that occur horizontally between such individual traders [11]. Investigating the horizontal relationships in supply chains is important to better understand actors' responses and associated outcomes such as vulnerability or effects on the environment [11,35]. Here, we study horizontal trade relationships in networks of traders of different types that have direct relationships with producers (often named patrons or middlemen). We call these networks regional trade networks. By traders we hereafter refer to any individual or cooperative entity that has stable relationships with fishers and engages in fish trading activities within trade networks (or that may even engage in fishing and trading themselves).

In this study, we address two specific research questions: 1) what trade-related processes take place in regional trade networks that are embedded in networks of social relationships; and 2) how do regional trade networks influence fish provision in situations of catch fluctuations, as found in spatially-heterogenous, multi-species fisheries? In particular, we investigate regional trade networks characterized by (i) certain proportions of trader types that are differentiated by their ability to buy fish from other traders [15], (ii) by certain degrees of overall connectivity, and (iii) certain degrees of connectivity across space.

To answer our research questions, we developed a multi-method approach, where we combine agent-based modeling, social network analysis, and qualitative data from a case study in Baja California Sur, Mexico. We designed the agent-based model (ABM) to investigate the trade networks through which traders interact in a small-scale fishery and to explore their importance for the stability of fish provision to regional markets. The model-building process was both informed by, and served to inform the analysis of qualitative case study data. This iterative process yielded insights on the relevant trading processes to implement in the model, thus addressing the first research question. In addition, the ABM was informed by literature and the analysis of an empirical trade network that yielded two model experiments. These experiments consist of empirically-informed network structures that differ in their connectivity patterns and proportion of trader types.

Through this combination of methods, we investigate trade processes within social network structures that are embedded in a social-ecological context. This allows us to analyze how traders' interactions across spatially-distinct regions and multiple fisheries influence fish provision under catch fluctuations. ABMs capture both micro- and macro-level processes and thus allow for the investigation of interactions between them [36,37]. We simultaneously investigate fish provision at two levels: at the system (macro) level, including fish provision to two different markets and the exploitation level of fish stocks; and at the micro level, through the effects of fish supply on livelihoods of individual actors (herein traders).

While choices about the design of the trading structures and dynamics in the model are based on analyses of qualitative and quantitative empirical data and knowledge from the case study in Baja California Sur, it is not a model of the case study. That is, social entities and processes are not parameterized using empirical data and the ecological model is simple and abstract. It should thus be interpreted as a stylized model that represents social and ecological processes in a realistic but abstract way, and with a focus on the details of the social structures and processes. This facilitates understanding the dynamics of spatial, multi-species and multi-market fisheries as driven by social processes and structures that can be found in regional trade networks such as the ones in Baja California Sur. Such fishery context can be found in many SSF around the world, even if the specific social processes and structures implemented could vary amongst cases and fisheries systems. Our modelling approach opens the door to future empirical research that may test the resulting hypothesis and investigate how similar processes and trade networks may operate in different case studies. Ultimately, through this multi-method approach, we provide guidance to specific research questions and factors that should be accounted for in future empirical research designs and fisheries policy discussions.

## 2. The Small-Trade model

We built an ABM named 'Small-Trade model' [38]. The purpose of the Small-Trade ABM is to understand how different patterns of trade interactions (i.e., trade networks) influence regional fish provision to supply diverse markets in a multi-species fishery that operates across multiple fishing regions. In this model, fishing and trading decisions occur between trading actors that interact at the micro level, ultimately affecting both system- and micro-level outcomes. These outcomes allow us to understand the dynamics of fish provisioning. System-level outcomes, such as the state of the fish stocks or fish scarcity at the markets, in turn affect fishing and trading decisions, as part of a process that is constrained and enabled by specific social and spatial network structures.

The model integrates empirically-based social structures (e.g., trade networks) and dynamics from the case of Baja California Sur (Mexico), with a theoretically-based ecological setting of different fish stocks and fishing regions. This section provides an overview of the entities and relationships in the Small-Trade model. The model processes are further specified in Section 4 based on results from the empirical analysis. See S1 Appendix for a more detailed description of the model following the ODD+D protocol.

There are two types of "agents" in the model: fish stocks and traders (described subsequently). Fishers are not represented explicitly because of their intimate relationships with traders [15,39,40]. This allows us to assume that traders have strong influence over fishers' fishing decisions and focus solely on the behavior of traders (see Section 4). Both traders and fish stocks belong to one of the two fishing regions specified in the model (Fig 1). These regions are not spatially explicit, but represent different fishing locations that provide fish to the same central point of commercialization (usually urban areas) in a stylized manner. We assume that the regions are geographically distant, so that there are no ecological interactions between them at the time scales of interest (i.e., weekly dynamics). Each fishing region has two fish stocks and the same number of traders (8 traders in each region).

There are four fish stocks (two per region, Fig 1) that have the same initial standing stock biomass and no connectivity between them (e.g., no trophic interactions). Each fish stock grows following a logistic growth function, reduced by harvesting that is proportional to fishing effort, according to the following equation: $S_{t+1} = S_t + rS_t\left(1 - \frac{S_t}{K}\right) - C_t$, following the Gordon-Schaeffer model, where $S$ is the stock; $r$, growth rate; $K$, carrying capacity, and $C$, the sum of all harvest extracted by traders. Each region has two fish stocks that represent two different finfish species (A and B) with the same growth rate and carrying capacity (Fig 1), representing an archetypical finfish stock based on data form the Gulf of California. While the two fish stocks are identical ecologically, they serve different markets (see below). We made this simplification to focus on market and trading dynamics and not dynamics that result from biological characteristics. See S1 Appendix for a detailed description of the fish stock dynamics.

There are two markets which represent the demand for the two fish species (Fig 1). For the purposes of this model, we assume that the aggregated market demand is constant and equal for the two species, and it is calibrated to a harvesting level (or fishing effort) that sustains each fish stock at maximum sustainable yield (MSY) during baseline model conditions. The two species (A and B) are differentiated by their market role and are not substitutable, therefore each type can only be sold to one of the two markets. Such examples of non-substitutability between fish species are common in fisheries [41]. For instance, in our case study, triggerfish (sold as processed fish in the regional market) and red snapper (sold as an entire fish to international and other high-value markets) are rarely substituted [42].

The social agents in the model are two types of traders: dealers and sellers. Dealers both buy and sell to each other, whereas sellers only sell the fish they obtain from the fishers [15]; this is

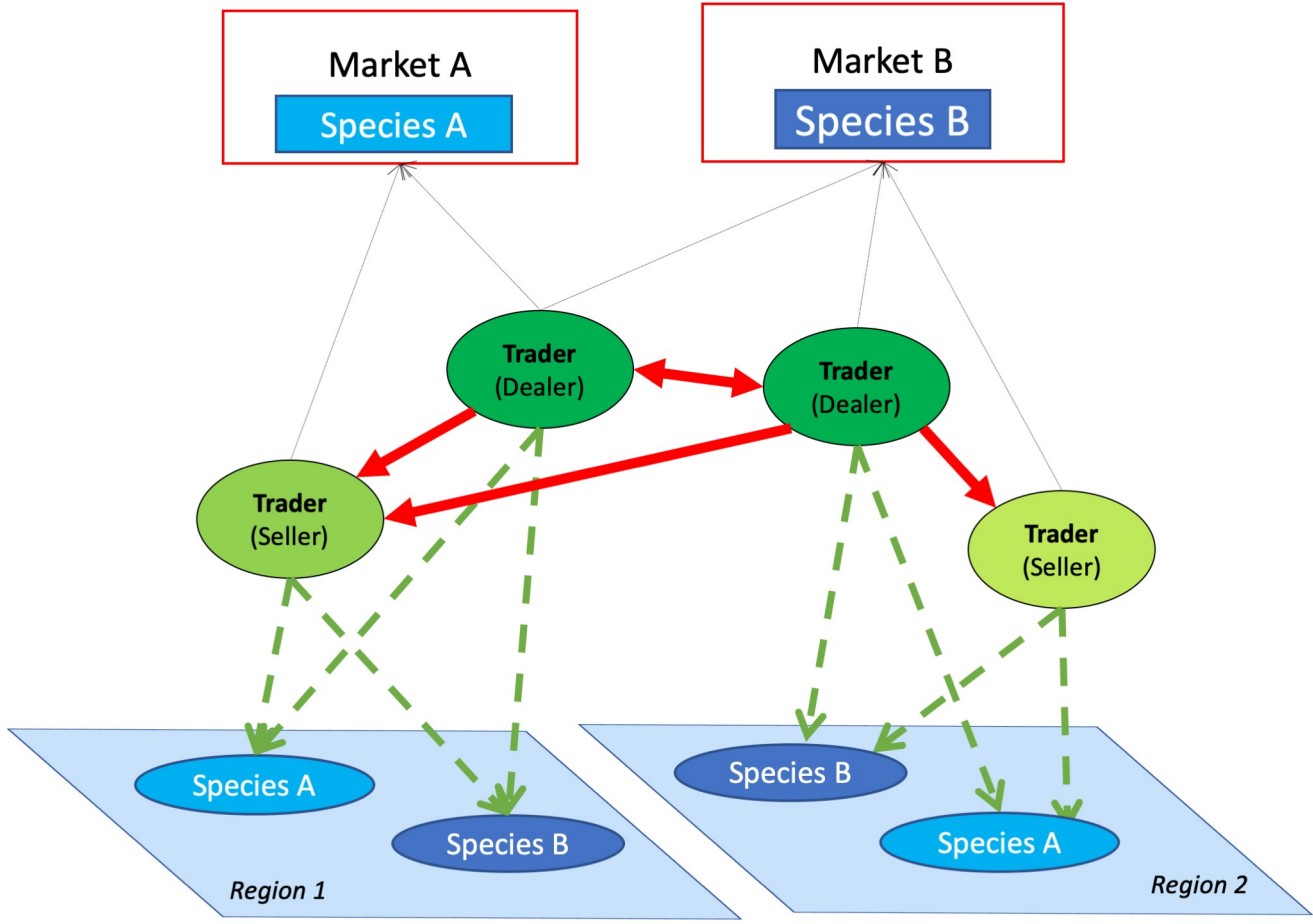

**Fig 1. Conceptual model representing the structure of the Small-Trade model.** The model aims to analyze a regional trade network (red links) between fish dealers and sellers, the two trader types that are connected to two final markets (red boxes) via vertical trade links (black links). Note that directionality of the red links (e.g. D→S) implies D "request fish to" and "buy from" S. Each trader is assumed to be connected to fishers (not explicitly represented) who allocate their fishing effort between the stocks of the two fish species within one region (A and B). Green lines represent these "fishing" relationships.

the only difference between trader types implemented in the model. Traders are connected by a trade network that allows them to sell to and/or buy fish from one another, and their main goal is to satisfy each other's demand for fish, which is the focus of this study (see Section 4). Trade relationships in the model can occur between traders targeting species in the same fishing region, or between traders in different fishing regions (Fig 1). To understand the role of this spatial dimension, we refer to the specific patterns of trade relationships between traders within or between regions as spatial connectivity. We assume that all traders are connected to both final markets, and thus, we refrain from elaborating vertical interactions in this model (Fig 1).

Each trader has access to the two fish stocks in their region, and is able to decide how to allocate their effort between them. In Baja California Sur, fishers and traders usually trade multiple fish species sold to different markets [43]. They can switch between species depending on species' availability (often seasonal) or the market demand (e.g., if there is an oversupply of one species, they can change to another one), among other reasons (as specified in Section 4). The effort allocated to each fish stock in the model affects both fish stock biomass and traders' supply of fish.

## 3. Analytical approach

The Small-Trade model is based on both qualitative and quantitative data from the case study of Baja California Sur, Mexico. The use of multiple sources of empirical data is common to parameterize agent-based models [44], including parameterization with empirical network structures [45]. However, in our case, we engaged in an iterative process where the building of the model, particularly its structure and rules, was informed by, and informed the analysis of empirical information collected through diverse methods (further described in Section 3.1). The analysis of an empirical trade network was used to generate empirically-informed network structures that were used as input to design model experiments (Section 3.2). Finally, we analyzed the model as indicated in Section 3.3 to understand how trade networks influence the dynamics of fish provision.

### 3.1. Assessing trading processes empirically

This study builds on qualitative and quantitative research investigating a finfish trade network in Baja California Sur in 2016 by González-Mon et al. [15]. In addition, a fieldwork campaign was conducted in May 2019 to investigate the dynamics of trade relationships. We conducted 13 semi-structured interviews (each approx. 1 hour in length) with finfish traders in Baja California Sur, after obtaining written informed consent and following the research ethics review process approved by the Stockholm Resilience Center's ethics sub-committee. Eight interviewees were among those interviewed in 2016, and the five others sourced fish from different regions in BCS not included in the previous study. This allowed for a broader understanding of the finish trading system in BCS, while also verifying and complementing previous information about the trade network collected in 2016 [15]. Interviews asked participants about the fish species they trade, trader's relationships with fishers and fishing communities, trader's responses to changes (such as events of fish scarcity, market gluts, or fishing closures), and their trading activity. Closed-ended (or survey) questions were also used to characterize the dynamics and nature of their trade relationships (see S2 Appendix).

The iteration between the conceptualization of the trading processes in the model and the analysis of the qualitative data, was key to answering the first research question and to identifying the key processes that characterize fish trading in the case study. The analysis of qualitative data often requires a process of abstraction and interpretation that can be facilitated by the modelling framework itself [46], and iteration between the analysis of empirical data and model simulation has been described as a useful process for understanding empirical phenomena [47,48]. In this way, the qualitative analysis of semi-structured questions consisted of thematic coding based on a preliminary conceptualization of the model processes. The resulting codes and themes (S2 Appendix), together with generalized findings from the literature, were used to inform the implementation of specific processes in the model (e.g., effort, decision-making, trading, as described in section 4). This analysis also helped design relevant model scenarios (Section 3.3), and ultimately, allowed understanding limitations of the model when compared to other empirical descriptions. In addition, the analysis of closed-ended questions provided qualitative patterns of the dynamics and characteristics of trade relationships, including how trade relationships were created and ended. This informed key assumptions of the Small-Trade model such as the decision to model stable networks (see Section 4).

### 3.2. Empirical network analysis and design of network configurations for model analysis

We used the empirical trade network data collected in [15]. This network represents trade relationships associated to finfish trade in the city of La Paz, the capital of the state of Baja

California Sur, from 9 fishing communities where fish is landed [15]. The data was collected through semi-structured interviews, participant observation and surveys with most of the fishers and traders in the aforementioned trading system [15]. In this study, we only included traders that have a direct relationship with fishers in the network (i.e., we excluded fish shops, intermediaries, and exporters). See S3 Appendix for more details. The analysis of this network allowed for further understanding of the network structure and the generation of model experiments.

We analyzed the network using the MpNet software [49] for exponential random graph modelling (ERGM). ERGM allows for the identification of potential mechanisms that explain network structures [50]. It is a probabilistic method that estimates the coefficients of certain network configurations (or "motifs") to fit a given empirical network [51]. Whereas understanding the mechanisms that explain the trade network was not the main question of this study, we used ERGM as a tool to generate simulated trade network structures built from the specific network-formation mechanisms (captured as different motifs) underlying the empirical network. This approach can generate any number of simulated networks with similar characteristics to the empirical finfish trade network in La Paz. In addition, it can generate hypothetical networks that differ from the empirical network in certain, controllable ways. We explored the influence of different network structures on the exploitation level and supply of fish using the Small-Trade model (see Section 2.3). Building on the empirical network (Fig 2A), we generated one

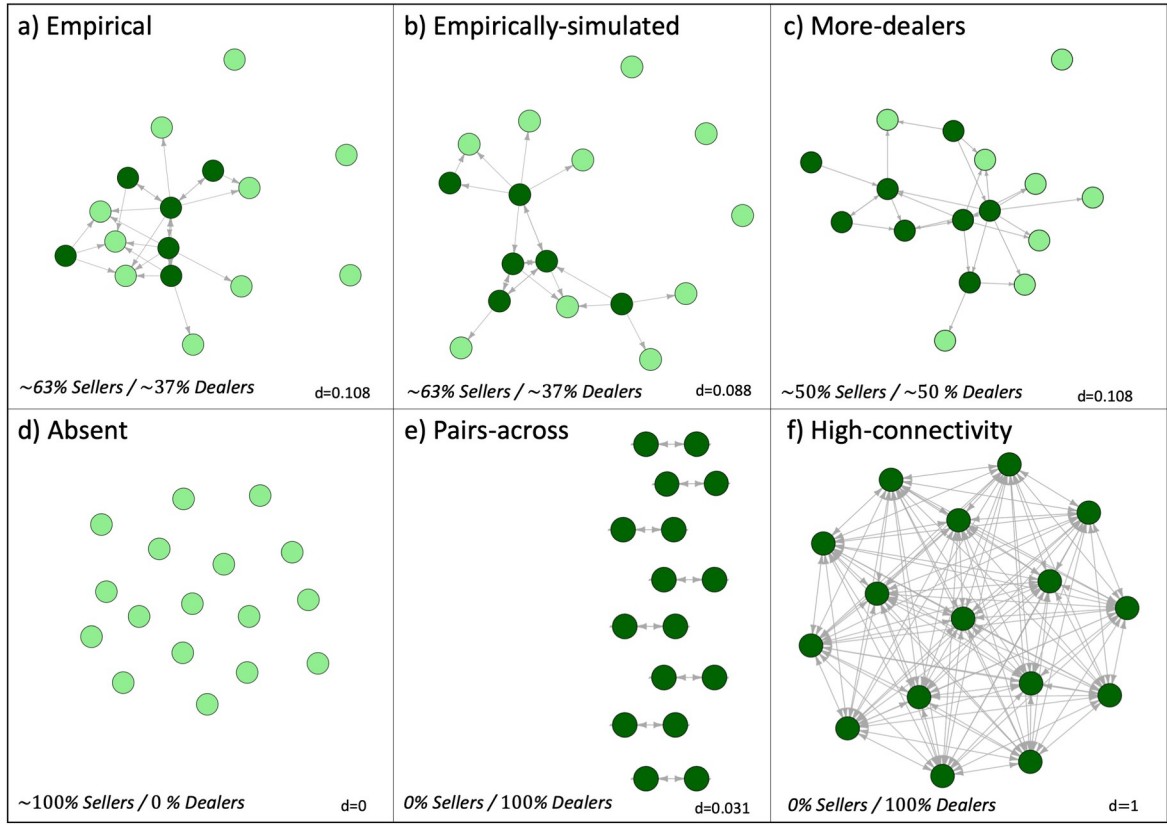

**Fig 2. Trade networks used in experiments for the Small-Trade model.** Color indicates different trader types (dark green, dealers; light green, sellers) and directionality indicates "buying from" or "requesting" (i.e. opposite of the flow of fish). a) Empirical network; b) Empirically-simulated network; c) network with more dealers. d- f are theoretical networks used in model experiments for reference: d) only sellers (absence of network); e) pair-wise network connecting traders from different regions; and f) fully connected network. NOTE: all networks have one added isolated trader for analytical purposes to have symmetry between regions in the ABM since the empirical network has 15 traders.

Empirically-simulated network (Fig 2B) that resembled the structure of the empirical trade network (in terms of maintaining the prevalence of the motifs under consideration). Following the empirical network, in the Empirically-simulated network, there was a significantly strong tendency for reciprocity in trade relationships (i.e., dealers buy and sell to each other more than would be expected by chance). There was also a positive effect of centralization of buying relationships (i.e., some dealers buy from more traders than others), even though this characteristic was not statistically significant in the empirical network. Second, we generated a hypothetical trade network that maintained some of the empirical network characteristics (i.e., number of links, average degree) but where more actors had the capacity to buy and sell fish and thus there was a higher percentage of dealers in the network (More-dealers, Fig 2C). See S3 Appendix for detailed results of the ERGM analysis and the network structures generated (Fig 2B and 2C).

To understand the influence of different trade networks, we compared the two empirically-informed network structures described above, and also compared them to three theoretical network experiments ("Absent", "Pairs-across network", and "Highly-connected network", see Fig 2). We explore two scenarios where we expose each of these networks to socio-environmental disturbances that affect fish availability (seasonality and catch variability scenarios in Table 1, see also Table 4 in S1 Appendix for details of the model specification). These environmental dynamics correspond to typical situations that can be found in a number of empirical contexts (in our case study and elsewhere). For every combination of a scenario and network, we performed 300 simulations to account for stochasticity. Thus, panel figures generally represent an average of 300 repetitions with the same initial conditions and parameter settings. In addition, we performed a sensitivity analysis with theoretical networks (Erdös-Rényi generated random networks based on the high-connectivity network in Fig 2F), varying the proportion of trader types and density (i.e., amount of links), in order to understand the effect of different isolated network characteristics on the model outcomes (see S4 Appendix).

### 3.3. Model analysis

We analyzed the emerging system-level outcomes of the model, as well as the micro-level dynamics of individual traders. At the system level, we measured fish supply in the two markets and the exploitation level of each fish stock (i.e., fish population). Fish supply is measured through fish scarcity (*market demand—market supply, >0*), and we included a measure of overabundance of supplied fish (*market supply—market demand, >0*, cf. fish waste) in supplementary results for reference (S4 Appendix), showing similar trends to scarcity. These measures allow us to understand the stability of fish provision by measuring short-term

**Table 1. Model scenarios.**

| Scenarios | Empirical example | Model interpretation |
| --- | --- | --- |
| **1. Seasonality** No catch or fishing of species A in region 1 | Species seasonal closure that can be associated to species' dynamics (red snapper stops "biting" in some regions certain times of the year, which interviewees referred to as when "the fishery closed itself"). It could also refer to fishing bans, such as a turtle protected area that banned fishing with specific gears. | Seasonal species catchability = 0 of species A during 6 out of 12 months in region 1, where individual catch depends on trader´s effort, fish stock level, and catchability of fish stock (see SM6 in S1 Appendix). |
| **2. Catch variability** High catch variability for all traders in both regions for all species | Catch fluctuates due to different environmental or social dynamics. | There is 20% stochastic variability in individual catch between the catch that is expected and the actual catch that is received. |

Scenarios of different catch dynamics implemented as exogenous changes in the model to investigate how the different network structures influence responses to such changes. Each scenario was tested for each of the network structures in Fig 2–2F.

fluctuations in fish supply. The stock exploitation level is an indicator of the sustainable exploitation of fish stocks. We present exploitation level as the percentage of the remaining fish stock relative to the stock that generates the MSY, according to the following equation: $\left(\frac{S}{S_{MSY}} - 1\right) \times 100$. Adding -1 means values below zero represent overexploitation, and values above zero represent under-exploitation. At the micro-level, we measure traders' average supply and supply variability, indicating the amount of fish they manage to sell (i.e., traders' fish supply excluding waste, either being fished or bought from other traders), including both species types. This measure is important for the stability of individual-level livelihoods and, in the long term, the potential of trading actors to remain in business. Here we may assume that traders' livelihoods improve with an increase in supply, as well as a decrease in supply variability (considering that fish is perishable and they have limited storage capacity). All outcome variables are measured as the average (median) across the last 5 years of the simulation (that runs a total of 20 years in weekly time steps).

In each simulation run, traders were allocated randomly between regions, maintaining equal proportions of the two trader types in each region. Thus, regions were equal in their fish stocks and trader types, but they could have different connectivity patterns within and between regions depending on the initial network setup. Regions could also be differently affected by exogenous drivers depending on the scenario (Table 1).

In order to understand the influence of spatial connectivity between traders and fish stocks, we measured the relative number of relationships within and between fishing regions using the external-internal index (E.I. index) in each simulation. The E.I. index [52] measures the embeddedness of a social network in differentiated groups, by comparing the number of relationships between actors of different groups (herein situated in different fishing regions) with the number of relationships between actors of the same group [53]. Therefore, we applied the E.I. index to measure the relative number of trade relationships (i.e. links) between the two regions in the model as compared to relationships within regions:

$$E.I. = \frac{(R1toR2Links - R1toR1Links)}{TotalLinks}$$

where $R1$ and $R2$ are the two regions in the model. Note that here, the E.I. index has been adapted to account for the directionality of links that is most relevant for the model scenarios. Therefore, it measures the degree of spatial connectivity of traders in region 1 (i.e., the R1toR2Links in the E.I. equation above), since seasonality is experienced by region 1 (Table 1). For each experiment, we analyzed the trade networks in relation to their E.I index across simulations.

## 4. Between empirics and modelling: Trading processes within networks

### 4.1. Key processes associated to regional trade

In summary, the modeled processes consist of: fishing; trading the fish catch between traders and selling it to the markets; communication between traders to request and assess the demand for fish; and a decision on how to allocate effort between species A and B before the next fishing event takes place (Fig 3; see Table 2 for a detailed description of the model processes). Each step in the model represents a week of fishing and trading activities, since there is usually a delay between fishing and trading due to the transport time and difficulty of accessing fishing communities in the case study [15,40]. Below we describe both the empirical and model interpretation of the two main processes included in the model: the trading strategies and effort allocation.

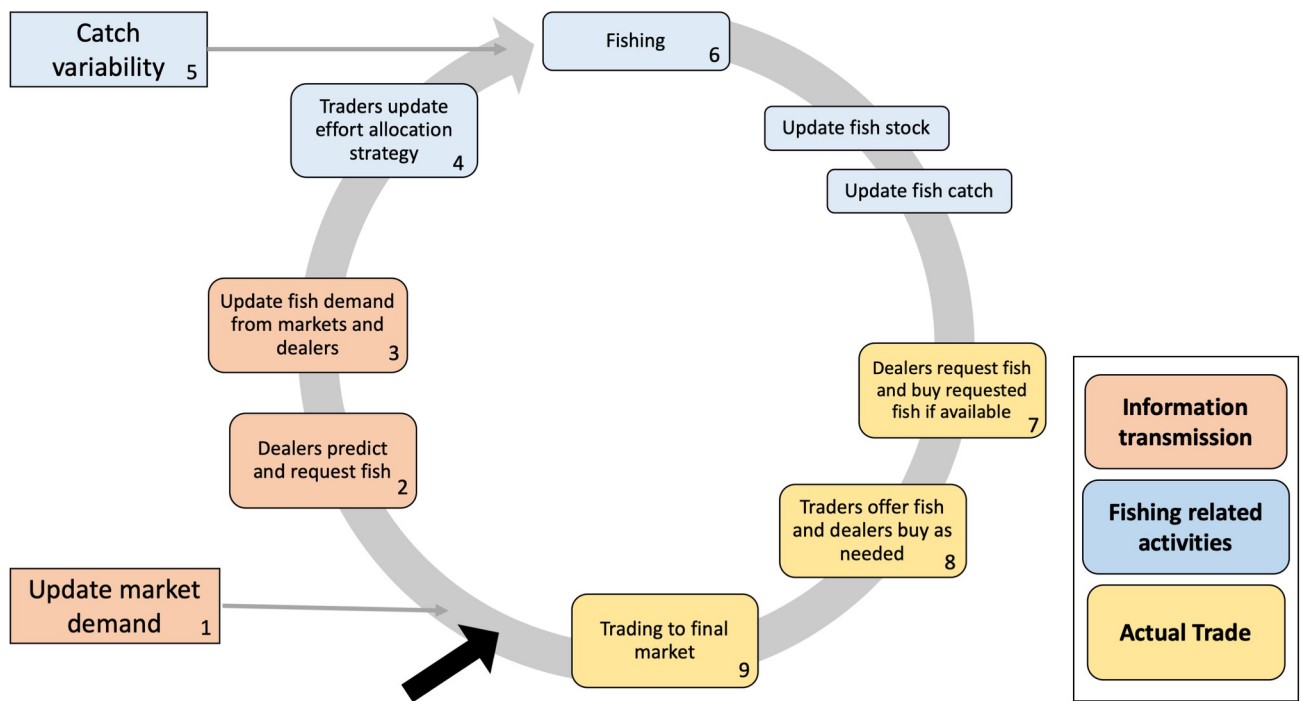

**Fig 3. Flow chart of the model process.** Two different types of activities are represented in the model: the social-ecological interactions between traders and fish stocks (blue); and trade strategies. There are two types of trade strategies: information transmission (orange); and actual trade consisting of buying and selling fish (yellow). The word "trade" refers to activities performed by both types of traders (dealers and sellers).

The trading strategies in the model consist of two interlinked types of social processes: i) transmission of information, represented by activities 1–3 (Fig 3) where traders request fish from each other, and take the requests from others into account to evaluate their demand (and ultimately the effort allocation); and ii) actual trade, represented by activities 7–9 (Fig 3), where traders buy and sell fish to each other and sell to final markets. The information transmission mechanism aims to represent how trading partners influence each other. In the model we assume that everyone has complete information about market demand, and that dealers are communicating equally among their network partners when requesting fish (Table 2). The actual trade mechanism consists of an activity where dealers reassess the request and offer an opportunity to all of their trade partners to sell them fish, followed by an activity where all traders have a chance to sell and buy until all trading opportunities are met (activities 7–8, Table 2). This two-step trade process represents a trading mechanism that consists both of requesting and offering fish to each other, since both actions were described in the interviews (see Table 2). In addition, activity 7 establishes an opportunity for trade between all partners in the trade network (representing stable relationships regularly used for trading), but activity 8 also allows traders to seize trading opportunities within their network (since, for instance, all traders may not have the same quantity of fish to sell). Finally, all traders sell to the final market until the market demand is met (and in case of oversupply, fish goes to waste).

Traders decide whether to increase their effort towards one of the species every week (with a maximum 20% increment), in which case they decrease effort for the other species, keeping the total effort constant (see activity 4 in Table 2). To model how traders decide how to allocate their effort, we developed a "combined" decision-making algorithm that accounts for 1) the demand for fish and 2) fish availability, based on empirical insights from the interviews with traders. Traders reported both influencing what the fishers catch based on their demand for

**Table 2. Weekly processes and trader activities in the model.**

| Key assumption | Empirical motivation | Model interpretation |
|---|---|---|
| **1.Update market demand** | | |
| Final market demand is not influenced by the trading activities in the regional fishery. Traders know the demand from the market. | Demand can come from markets at other scales. See for example [54,55]. | Exogenous constant. Each trader updates their market demand based on this value. Calibrated: calculated as being equal to the catch at MSY. |
| **2.Dealers predict and request fish** | | |
| Dealers have expectations on their need for fish and, if they cannot meet the demand through their own fishers, they ask other traders with whom they have stable relationships. All relationships are equally committed. | Traders can be in communication with the fishers to know how much fish they are going to get, and can call their buyers to know how much fish they need*. Traders report knowing who to ask if they need fish*. | Traders use the market demand, the demand they had from their trade partners, and catch they obtained to predict and request fish from others. Traders request when they could not meet their demand with their own catch in the previous time step, and request equal from all their trading partners (out-links). |
| **3. Traders update demand from dealers** | | |
| Traders communicate between each other and try to meet each other's demand for fish. | Traders are in communication with their stable trading partners and "know" what they need to buy and sell in advance*. | Traders update their demand for fish based on what the dealers request to them. |
| **4. Traders decide/update effort allocation between species** | | |
| Traders can influence fishers' target species, and decide the need for fish based on their market demand and fish catch. However, they cannot change the total effort. | Traders report requesting fishers the species they need or not, or even stopping to receive certain species altogether if they don't need them*. Traders also report that fishers fish what they can based on what is available or what they want based on their preferences*. | "Combined" decision-making algorithm: traders balance equally (at 50%) their relative need for each fish species with the relative Catch Per Unit Effort, and change their effort allocation between species accordingly (with a maximum 20% increment). Effort is constant. |
| **5. Catch variability** | | |
| Catch can be affected by factors other than fishing effort. Fish availability and catch is variable and influenced by multiple factors related to the ecology, biophysical and institutional conditions. | Fish catches fluctuate and fish species are not available all the time at all places*. See also [56–58] describing examples of catch variability and change in this case study. | Represented by two variables 1) Individual catchability ($ic$), a trader variable that changes each week to influence a trader's individual catch ($ic$ is normally distributed with mean = 1 and SD = 0.2) and 2) seasonal species catchability ($sc$), a species variable that changes seasonally depending on the scenario (see Table 1). Traders catch is then calculated as $effort * stock * ic * sc$ |
| **6. Fishing** | | |
| Traders have some capacity to influence fishers' effort allocation strategy. Traders ask their fishers for fish and fishers go fishing following traders' advice. | Some traders report influencing what species fishers catch, even if this is not true for all traders*. Traders' influence on fisher's decision-making has been reported across several case studies [24,26,59]. | Traders go fishing, influencing the fish stock (that regenerates according to a logistic growth function) and obtaining catch that depend on their fishing effort, the stock level, and the catchability of the stock. |
| **7. Dealers request and buy fish based on their needs** | | |
| Traders have moral commitment between each other, and they try to sell/buy to all their trading partners to maintain their relationships. | Traders report calling each other to request what they need, and some traders try to trade regularly with all their stable trading partners*. Strong network relationships are used frequently [15,18]. | Dealers evaluate their fish demand and request an equal amount of fish to all their trade partners. First sellers sell to dealers according to that request, and then dealers sell each other according to that request. |
| **8. Traders offer fish and dealers buy based on their needs** | | |
| Traders can offer fish to their stable partners based on opportunity. Dealers will buy the fish they need to satisfy their demand. | Traders report calling each other to offer fish (mainly suppliers/sellers call offering the fish they have) *. | Traders offer their remaining fish to their trading partners at random, and dealers buy if they still need more fish to meet their demand, until all possible demands within the network are met (or there is no fish to sell). |
| **9. Traders sell to the final markets** | | |
| Traders sell to the markets, which have a limited demand. | Traders sell to different fish shops, restaurants, and exporters within the state, or even export fish themselves nationally or internationally [15]. | Traders sell their catch to the markets (buyers) one by one activated at random. When the markets are full, fish goes to waste. |

The table provides details on the main assumptions, relationship with primary and secondary empirical information, and model interpretation. Note that the same process applies to both types of fish stocks.

*Results from the qualitative analysis, see S2 Appendix for more details on the themes informing the narratives presented.

fish, and not being able to control what fish is "biting" at the moment or what fishers fish (see S2 Appendix). Thus, the empirical data does not allow for the definition of one single way of allocating effort between species. Therefore, the combined decision-making algorithm balances two different decision-making sub-models: i) a demand-driven model, which assumes that each trader's objective is to meet the need for fish of the market and their trading partners (aiming neither to over supply or under supply species A or B); and ii) a CPUE-driven (catch per unit effort) model, which assumes that each trader's (and fisher's) objective is to maximize their CPUE for each species. This decision occurs weekly, and is specified according to the following decision-making algorithm (see SM5 in S1 Appendix for details):

$$\text{If } de_A > pe_A \text{ then } e_A = de_A - mag_A$$

$$\text{If } de_A < pe_A \text{ then } e_A = de_A + mag_A$$

$$\text{If } de_A = pe_A \text{ then } e_A = de_A$$

where

$$mag_A = br \times abs\,(de_A - pe_A).$$

Where $de_A$ denotes the optimal effort based on meeting the demand for species A according to the demand-driven sub-model; $pe_A$, the optimal effort based on maximizing the CPUE for species A according to the CPUE-driven sub-model. The influence, or weight, of each sub-model is determined by a balance rate (br, with range from 0 to 1). $mag_A$ represents the magnitude of the effort difference between the demand versus the CPUE optimal efforts. $e_A$ denotes the new balanced effort for species A. This decision-making algorithm is repeated for species B, where $e_B$ denotes the new balanced effort for species B. The final effort is then balanced between the species in relation to E:

$$e_{A\_new} = \frac{E}{(e_A + e_B)} e_A$$

$$e_{B\_new} = \frac{E}{(e_A + e_B)} e_B$$

where $e_{A\_new}$ and $e_{B\_new}$ are the actual efforts applied for species A and B respectively, and E is the maximum effort that the trader can apply and need to share between species A and B. Following this combined decision-making algorithm, if the balance rate puts more weight into the demand-driven decision-making (br<0.5), fishing effort is allocated to one species even if it is not available (when seasonal species catchability = 0). This is not sensitive to findings that suggest that fishers will not target species that are not "biting" (S2 Appendix). On the contrary, when the balance rate puts more weight into the CPUE decision-making (br>0.5), the more abundant species are systematically oversupplied. This process is not sensitive to the existence of a limited market demand where traders may not buy what they cannot sell (S2 Appendix). This study uses a balance rate that puts equal weight into both decision-making sub-models (br = 0.5), and provides a sensitivity analysis to evaluate the effect of this parameter (S4 Appendix). Note that individual factors influencing fishers' decision-making, such as preferences or culture, are outside of the scope of this study (see e.g., [60,61]).

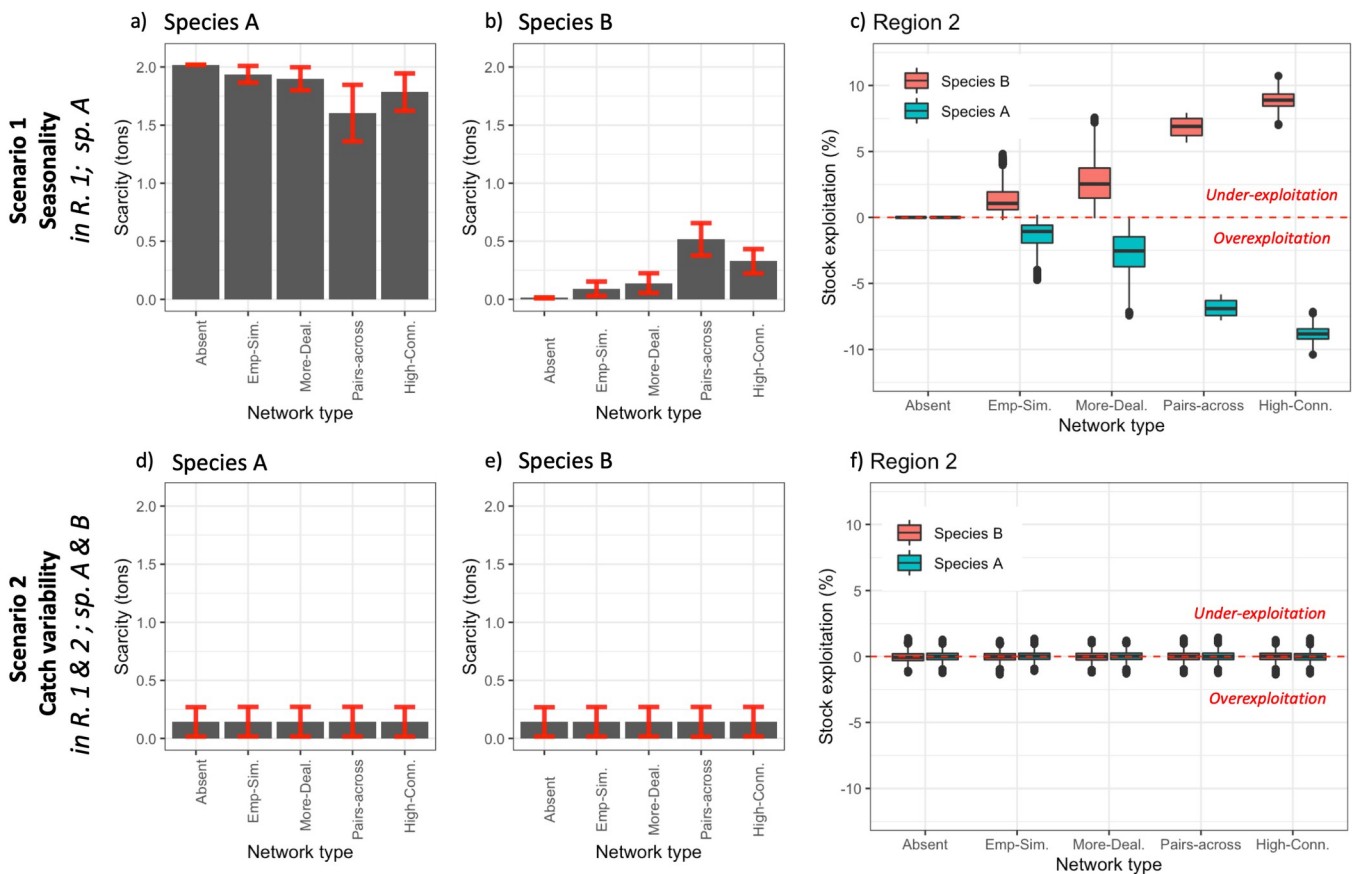

**Fig 4. Influence of network structures on fish supply and stock exploitation.** Panels a-c show scenario 1, the seasonality scenario (seasonality of species A in region 1). Panels d-f show scenario 2, the catch variability scenario. Panels a, b, d and e show the average (grey bars) and variability (standard deviation, red interval lines) of fish scarcity in the two markets (for species A and B). Panels c and f show the distribution of the exploitation of the two fish stocks in region 2 as compared to the reference point of exploitation at MSY (dotted red line). Note that the exploitation of fish stocks in region 1 is not affected by the network structures (see S4.2 and S4.6 Figs in S4 Appendix). Network types following Fig 1, and ordered from less to more dealers and connectivity. Results shown are averages and variability across 300 runs for the last five years of simulation.

## 4.2. Model verification: The fish request mechanism can generate spillover effects across regions

Without seasonality or catch fluctuations in the Small-Trade model, there is no fish scarcity and fish stocks are exploited at MSY. When there is a seasonal "closure" of species A for traders from region 1 (seasonality scenario, Table 1), regional trade networks trigger changes in fishing effort among the traders from region 2. The model shows that traders from region 2 increase the effort of species A in that region to supply fish to traders in region 1 experiencing the fish closure (i.e., when there is lack of species A in region 1). As a consequence, trade networks decrease scarcity in market A (Fig 4A), but the opposite occurs for market B (linked to the non-seasonal species) where there is an increase in market scarcity due to changes in effort allocation (Fig 4B). In relation to this, there is overexploitation of the seasonal species (A) and under-exploitation of the non-seasonal species (B) in region 2 (Fig 4C). This is due to requests placed during the information transmission strategy and the (partially) demand-driven decision-making algorithm that assumes traders aim to satisfy each other's need for fish. In this case, traders from region 1 request more of species A due to the closure. Traders from region 2 will then try to satisfy their request if they have a relationship with each other (i.e., when

connected through a regional trade network). However, in the absence of such regional trade, traders (in region 1) change their effort allocation in response to the closure of species A, which initially leads to a slight overexploitation and oversupply of species B in region 1 (S4.11 Fig in S4 Appendix), but these changes do not affect region 2 (Absent network in Fig 4C). Therefore the trade process implemented in the model generates cascading or spillover effects across space triggered by changes in fish availability in one region, confirming that the model works as expected. These results are further detailed below for all network structures analyzed.

## 5. Dynamics of fish provision in response to catch fluctuations

### 5.1. The number of dealers and their connectivity influence fish provision when dealing with seasonal variability

Networks with a higher number of dealers have a stronger effect on fish provision; for example, a higher number of dealers reduces the scarcity of fish in the market but also increases overexploitation of the seasonal species A (Fig 4A–4C; S4.1 Fig in S4 Appendix). The More-dealers network (Fig 2C) decreases market scarcity of the seasonal species to a greater extent than the Empirically-simulated network. The More-dealers network also has a stronger, opposite effect on the non-seasonal species (Fig 4B). This stronger effect is also generally true for the overexploitation dynamics, due to larger changes in effort allocation between the two species (Fig 4C).

As an exception, the Pairs-across theoretical network is able to reduce scarcity at market A more than the Highly-connected network (with the same number of dealers but higher connectivity). This was found to be the case even when the Pairs-across network generates a lesser increase in overexploitation (Fig 4A–4C). Thus, when there are only dealers, higher connectivity does not necessarily mean less scarcity of the seasonal species (S4.4 Fig in S4 Appendix). An overly-connected network, such as the Highly-connected network, can increase scarcity of the seasonal species and increase its overexploitation, when compared to networks with lower connectivity (Fig 4A and 4C). In this case, the high number of relationships implies that each of the traders gets less accurate information about other traders' needs, since the fish request is split over several traders. As a consequence, there is a smaller change in individual fishing effort. This implies that traders from region 2 do not increase their effort of the seasonal species A as much when there is a lack of that species in region 1, which results in higher overall scarcity of the seasonal species when it is closed for fishing in region 1. It also implies a higher fishing effort over the seasonal species in region 2 when it is being fished and producing a higher yield in region 1 since it is under-exploited in that region due to the effect of the closed season (see S4.12 Fig in S4 Appendix). This impedes the recovery of the stock level of the seasonal species in region 2 during the periods of oversupply of that species in the markets. On the contrary, the Pairs-across network represents an extreme case with low connectivity (only one link per trader), where the relationship always goes to a trader in the opposite region. Thus, each trader in that network satisfies the needs for fish of a trader in the other region when a species is closed or overexploited in one region but not the other. This implies larger changes in individual fishing effort that respond quicker and more accurately to trader´s requests.

In addition, the outcomes of the trade network experiments are influenced by the type of environmental dynamics that affect the system, as revealed when comparing the seasonality scenario with a scenario where catch varies randomly. Trade networks influence fish provision when the variability affects only one region (seasonality scenario, Table 1), as trade networks can "buffer" local environmental changes by changing fishing effort elsewhere, as explained above. However, when both regions are affected by random variability in fish catches (catch

variability scenario, Table 1), regional networks do not seem to play a role in fish provision at the market level, nor in the exploitation dynamics of fish stocks (Fig 4D–4F). Note that in this scenario, trade networks do affect individual-level outcomes (see section 5.3), and therefore, they influence fish distribution amongst traders but do not seem to have a significant effect on fishing effort due to the random variability in fish catches.

## 5.2. Spatial connectivity increases fish provision and spillover effects

When analyzing the spatial connectivity patterns of the Empirically-simulated and the More-dealers network, the Small-Trade model shows that an increase in connectivity between regions also leads to a stronger effect on the system-level outcomes associated with fish provision, in response to seasonality in one region (Fig 5). The more connectivity between regions, and particularly the more request links from region 1 to region 2, as compared to relationships within region 1 (higher E.I. index, Fig 5), the less fish scarcity in market A and the more fish scarcity in market B (Fig 5A and 5B). Higher connectivity between regions also leads to more overexploitation of species A and more under-exploitation of species B in region 2 (Fig 5C and 5D). This qualitative pattern holds for both network structures (Fig 5). The differences in the average scarcity and exploitation observed between the Empirically-simulated and the More-dealers network (Fig 5), correspond to the differences mentioned above (Fig 4A–4C), but they highlight the influence of spatial connectivity.

Note that the exploitation dynamics in region 1 are not affected by the network structure or the connectivity between regions (Fig 5E and 5F). Given that in scenario 1 seasonality only affects region 1, stocks in that region are mainly affected by changes in effort allocation within the region. Therefore effort is mostly influenced by the closed season of species A; it is not as influenced by the network connectivity and the request for fish of traders in other regions. This is a consequence of the assumptions associated with the effort allocation algorithm (section 4.1), and we acknowledge that another response that could be expected is an increase in effort for the non-seasonal species (B) when the seasonal species (A) is closed; this would increase overexploitation of species B in region 1 and decrease the scarcity of species B to some extent.

## 5.3. The more buying relationships the more stable provisioning for traders, unless they are too connected

Regardless of the type of environmental change affecting the system, trade networks generally increase the dealers' fish supply as compared to the absence of networks, such that dealers with more buying relationships (higher outdegree) increase their fish supply (Fig 6A and 6C). In addition, higher outdegree would most often decrease traders' average supply variability (Fig 6B and 6D). Sellers, who have no buying relationships, have the lowest average supply and highest supply variability for most networks (Fig 6). However, increasing the number of relationships does not always increase dealers' supply (or decrease their supply variability) significantly, since we observe a plateau effect for the higher outdegree centralities in both the More-dealers and the Empirically-simulated network (Fig 6). In addition, the results of the Highly-connected network indicate that there is not a significant increase in traders' supply when they have an extremely high number of connections in a network with only dealers (Fig 6). Overall, the average supply and supply variability seem to be related to the position of the dealer in the network (i.e., to their outdegree centrality), in both the More-dealers and the Empirically-simulated networks (Fig 6).

Despite similar patterns of the influence of the number of relationships on traders' outcomes in both scenarios of environmental change (scenarios 1 and 2), we observe some qualitative differences (Fig 6C and 6D as compared to Fig 6A and 6B). For instance, in the catch

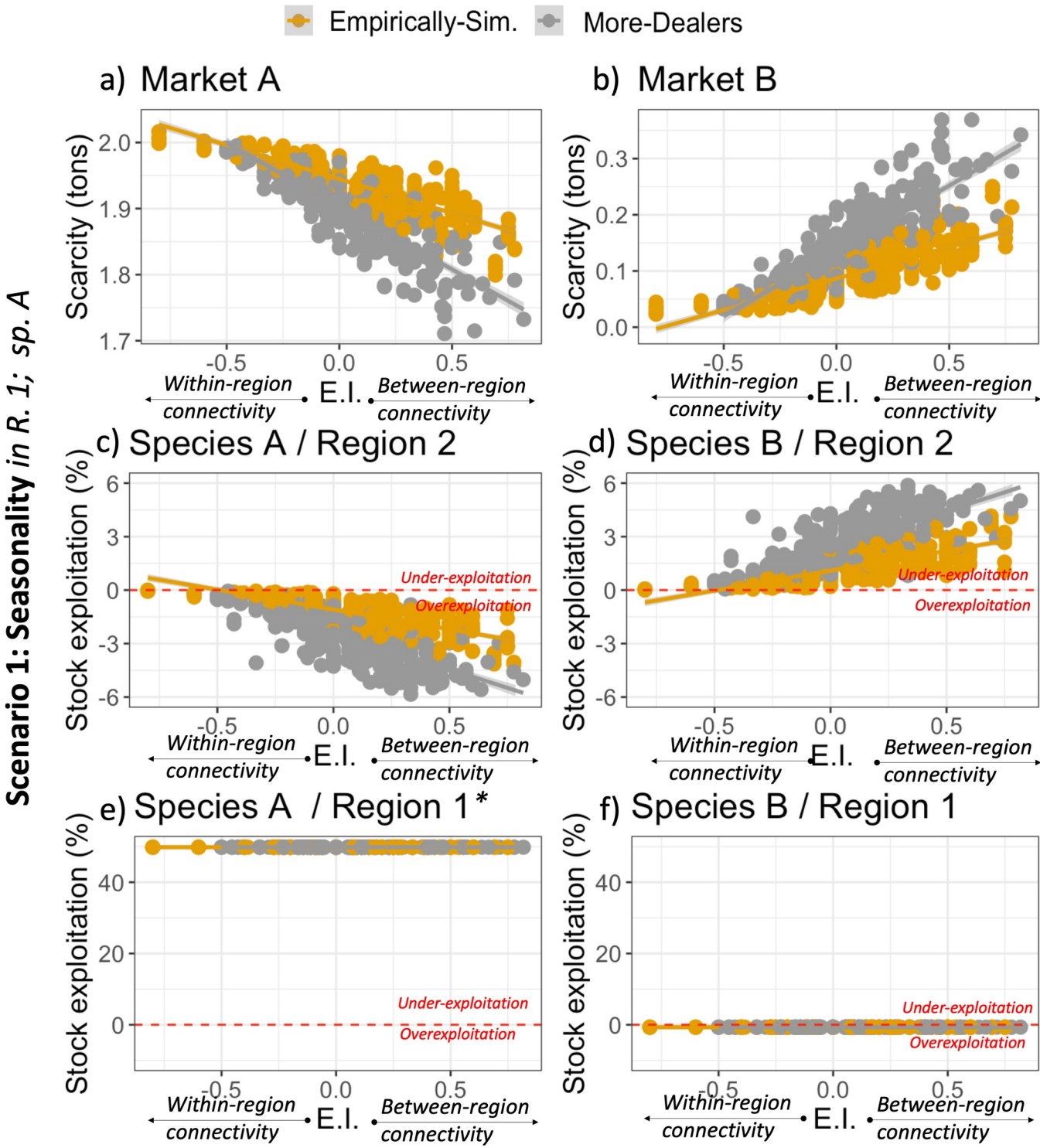

**Fig 5. Influence of connectivity between regions for scarcity and stock exploitation in response to seasonality in Species A in region 1.** The E.I. index represents the spatial connectivity of traders in region 1, where EI>0 implies more request links from region 1 to region 2 than within region 1; and EI<0 indicates more requests links within region 1 than between the regions. Panels a-b show average scarcity, and panels c-f show the average exploitation index of the four fish stocks as a percentage, compared to the reference point of exploitation at MSY (dotted red line, where <0 represents overexploitation). Network types following Fig 1. Note that the scale of the y-axis changes between panels (panels cannot be quantitatively compared). Results show 300 simulations per network type, where each dot is the average in one simulation. *The high under-exploitation of the seasonal species (A) in e) is because it is not harvested during half of the year due to an external constraint imposed in the scenario (see Table 1).

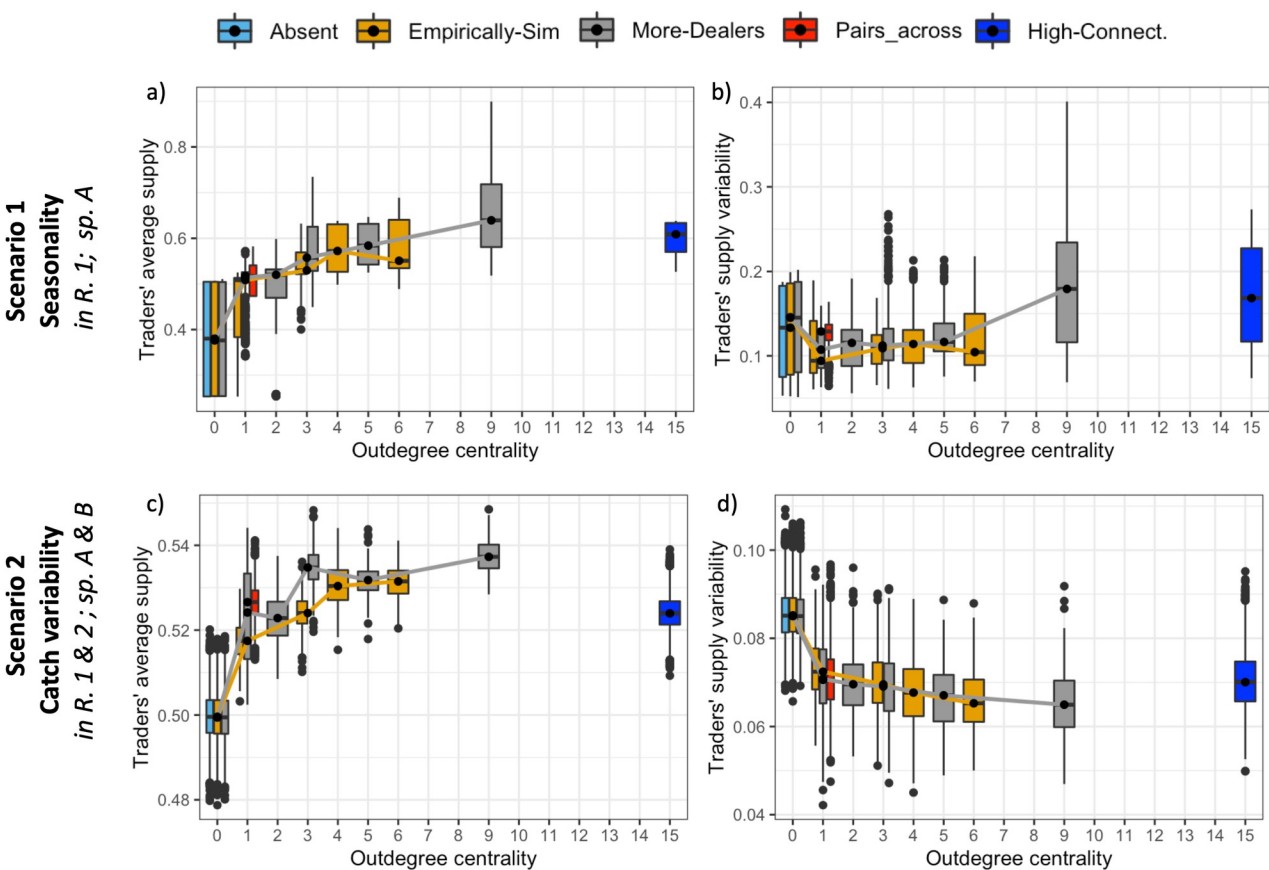

**Fig 6. Influence of traders´ position in the network on individual traders' supply.** Panels a-b show the seasonality scenario (seasonality of species A in region 1). Panels c-d show the catch variability scenario. Outdegree represents number of outgoing links, where out-going means requesting to or buying from, and thus sellers have outdegree = 0 by definition. Box and whisker plots represent median (black line in the box connected by colored lines per network type) and variability (quartiles represented by the box and whiskers) in traders´ average supply (a and c) and in traders´ supply variability (b and d), calculated across the 16 traders in 300 runs per network type. Dots are outliers. Network types following Fig 1, represented in different colors. Please observe that direct comparisons of degree centralities across network types should be done with caution since the networks differ in respect to several factors. This caution does not apply when comparing degree centralities within the same type of networks.

variability scenario, most networks increase dealers' fish supply and decrease their supply variability as compared to sellers, and the absence of regional trade networks (Fig 6; S4.3. and S4.7 Figs in S4 Appendix). However, when facing the seasonality scenario, dealers with a high number of relationships increase their fish supply, but this comes at the cost of also increasing their supply variability, as evidenced in the More-dealers network and Highly-connected network where some dealers with more than eight buying relationships had increased supply variability (Fig 6B). The tradeoff of increasing supply at the expense of increasing individual supply variability in the seasonality scenario, as opposed to the catch variability scenario, also holds in other theoretical networks analyzed (see S4.3 and S4.7 Figs in S4 Appendix).

## 6. Discussion

### 6.1. The key influence of regional trade networks on the dynamics of fish provision

The Small-Trade model shows that regional trade networks can under some conditions decrease fish scarcity at the final markets and increase the stability and supply of fish for some traders. However, this can also create spillover effects between regions and species in response

to local differences in the size of fish stocks and their catchability (or accessibility). In this way, the seasonal closure of one fishery (seasonality scenario, Table 1) can lead to overexploitation and changes in the market supply of other species. We show how such spillover patterns, that are related to sequential exploitation or displacement of fishing impacts, can result from two simple interacting processes in a socially-embedded regional trade network. This contributes to the literature showing similar patterns of sequential overexploitation associated with global trade (see e.g., [29,62]).

The first process relates to a trading strategy where some traders (dealers) can request fish from traders operating in regions where fish is more abundant, and where the social embeddedness of traders provides a motivation for all traders to satisfy each other´s need for fish (for example, in order to maintain their trade relationships). This implies that trade relationships are characterized by a certain level of commitment or trust between traders (Table 2), which has been previously described in case studies of SSF across the world [8,30]. This trading process contrasts with common conceptualizations and models of trade that are based on strict market principles and price-making or bargaining mechanisms. We purposely investigate the influence of trading strategies that, albeit not contradicting the existence of market mechanisms, are nonetheless mediated by social processes as evidenced by our interview data. Still, we acknowledge it is only a limited representation of all the economic and social processes that potentially influence trade. The second process in the Small-Trade model builds on traders' ability to allocate effort between different species. Diversified fishing strategies that imply changing the allocation of fishing effort are important for SSF around the world [59,63,64]. Here we emphasize that not only may fishers diversify their target species, but also certain types of traders that have a close interaction with fishers and fish production activities. This suggests the need to account for interactions that depend on social relationships and species diversification when designing research and management programs that aim to influence SSF value chains.

Diversified trading strategies (e.g., maintaining a diversity of trade relationships) that buffer changes in supply to satisfy final markets, are also used in global and regional trade networks [28,34]. In this context, the specific patterns of spatial connectivity between the trade networks and the two fishing regions can be key to understanding how trade networks can respond to different environmental changes (cf. [65]). For instance, our model suggests that trade relationships between different regions are especially important when the regions are not facing the same type of change in fish availability or are subject to different ecological dynamics (Fig 5). Similarly, our results also suggest that when regions are facing the same type of changes, cross-regional trade relationships do not affect system-level fish provisioning in any way if resource conditions are equal. Notwithstanding this finding, trade relationships within the same region, and/or across regions, can be important and increase individual traders' capacities to deal with changes such as catch fluctuations, regardless of whether changes are localized or global, as hypothesized by González-Mon et al. [15] and as evidenced in this model (Fig 6). Therefore, in order to better understand the influence of trade networks on the dynamics of fish provision, there is a need for future empirical research that explicitly investigates the spatiality of trade networks, and considers the various types of ecological and social dynamics that can manifest differently across places.

Research that accounts for these complexities is especially important since the dynamics highlighted by the Small-Trade model could have implications for the fisheries of Baja California Sur, as well as other SSFs. Environmental dynamics such as changes in sea water temperature and hypoxia events [66,67], and institutional dynamics such as those resulting in new local seasonal closures [68], are likely to change local species availability. In this context, the existence of dealers and horizontal relationships in trade networks could have an important

role to play in guaranteeing fish provision and traders' livelihoods. However, the increased stability and supply of fish may mask environmental dynamics and signs of stock overexploitation that could threaten the long-term capacity of the system to provide fish [24,27,69]. In addition, an increase in dealers and the connectivity of trade networks—for example, due to the development of infrastructure (e.g., roads) or interventions aimed at increasing the market capacities of fishers and fishing organizations—could increase actors' capacity to deal with changes and provide food, but also enhance masking and spillover effects, potentially leading to unintended consequences. Thus, a systemic assessment of the potential trade-offs and consequences of trade networks and different trading capacities could aid in the understanding of the future development of small-scale fisheries, such as those in Baja California Sur.

## 6.2. The role of trader types and trader connectivity

We show that the positions of dealers in trade networks influence how horizontal relationships can increase their individual capacities to deal with change. Empirical studies have shown that traders perceive varied benefits from seafood trade [30], and previous work has also suggested that more central actors in the network (e.g., with higher number of relationships) have higher capacities to deal with changes ([15] and references therein). Considering the trading strategies included in the model, we show that more central actors tend to have higher fish supply and lower supply variability on average, which could suggest an increased capacity to deal with changes. However, we also show that this is not always the case, since over-connectivity can also lead to higher supply variability for traders and, at the system-level, overconnected networks can increase overexploitation while not increasing fish supply at markets. In our model, a fully-connected network implies the transmission of less-accurate information about the needs of each trading partner as compared to pair-wise relationships, which could resemble a hypothetical situation of trading with every trader rather than through more committed personal relationships. This finding resonates with seminal empirical studies on the embeddedness of trade networks (cf. [70,71]) that suggest that an intermediate number of embedded relationships leads to better outcomes at the individual and system-level. It also relates to research on information networks, highlighting that increased connectivity at the individual level can increase actors' performance (as observed here in section 5.3), but high connectivity overall may not lead to better outcomes at the system level [72].

In the context of a recent trend towards intervening in fisheries value chains for fisheries sustainability [73], we argue that promoting changes in market structure may have consequences for the capacity of traders and trade networks to deal with changes influencing fish provision at the individual and system levels. Such development interventions should be aware that a certain level of horizontal connectivity may be beneficial for some people and at certain times. For instance, promoting only vertical interactions and neglecting horizontal connectivity could threaten actors' capacities to deal with change, but increasing connectivity for all actors may also come at a cost. However, this remains a hypothesis to be tested by future empirical research in the context of fish trade networks. In particular, future studies could investigate the specific influence of traders' connectivity and their network position on their capacities to sustain fish provision in face of different social and ecological changes. Research comparing different types of traders in terms of their relationships with others, as well as comparing network structures with different characteristics, could further advance the hypotheses arising from our model. In addition to the trading strategies and outcomes explored in the Small-Trade model, investigating actors' potential influence in relation to their network position would require insights into traders' agency, capacities, and motivations to steer the development of the fisheries towards specific objectives [15,19].

### 6.3. Future model extensions and open research questions

The Small-Trade model combines network simulation and analysis with agent-based modelling to investigate the interplay of spatial connectivity of traders with institutional and ecological dynamics of the fishery. This allows us to explore the implications for fish provision at the individual and system levels. Combining agent-based modelling with network analysis has recently been highlighted as an important methodological development for social-ecological systems modelling, to which we contribute [45,74]. Future extensions of the model may address how different types of traders or market dynamics can increase pressure on specific species that are connected to international markets or high demand, and situations where different types of traders can have differential access to markets or market concentration. Research could also address the influence of higher heterogeneity among traders, who may have different capacities or behaviors regarding their trade strategies, their effort allocation, or their relationship with fishers. There is also a need to understand how multiple species and regions interact through ecological processes, and how changes in the market preferences or substitutability of different species could influence both ecological and social outcomes.

We acknowledge that other trading mechanisms or strategies not included in this model could increase individual-level fish supply, and potentially influence system-level outcomes. For example, understanding the role of infrastructure to freeze catch could be an interesting addition to the Small-Trade model, in a context where supply chain interventions are often targeted towards improving post-harvesting processes by increasing access to freezing facilities (e.g., [8]). The use of freezing facilities would allow some actors to store fish and sell it in times of scarcity, which would decrease fish waste and scarcity at the system level and allow traders to better deal with catch fluctuations [8]. Thus, the trade-off between increasing average supply and supply variability of traders (Section 5.3) may not be as detrimental for actors who own freezing facilities; instead, it could constitute an opportunity to store fish during periods of abundance and use it during periods of scarcity. In our case study, there is limited processing and freezing infrastructure, but several traders who buy from others (dealers) own a warehouse and a few have freezing facilities, which would influence fish provision at the individual and system level.

In other respects, a common way to deal with oversupply (and avoid waste) is to select new buyers through more temporal (or weak) trade relationships [18], which are also observed in Baja California Sur ([15]; S2 Appendix). The strategy of maintaining both committed or stable relationships and sporadic relationships (i.e., both strong and weak ties) may play an important role in providing flexibility in the supply chain [18,71,75], and therefore contribute to the stability of fish provision. These processes are not considered in this study, but future model extensions may better capture differential and more adaptive requests between trading partners, and more adaptive behaviour in choosing who to trade with, which are indicated by empirical data. In this context, mechanisms related to loans (of fish and/or money) and the exchange of financial services can coexist with trading strategies (e.g., [20]), leading to changes in network relationships at longer time scales. Inclusion of the capacity of actors to create and end trade relationships would be an important addition to the model; it would allow for simultaneous consideration of the role of trade networks for fish provision and endogenous processes that may change network structures influenced by (and influencing) fish provision [76,77].

## 7. Conclusion

The Small-Trade model highlights that traders that have direct relationships with fishers and can also buy from other traders (referred to as "dealers"), constitute "horizontal" trade

networks, where the number of dealers, as well as their position in the trade network, affects fish provision. In referring to horizontal networks, we emphasize that these trade relationships occur between traders at the same stage of the value chain (i.e., those directly buying from fishers), in contrast to value chain studies that focus on vertical relationships within the chain. Several previous case studies have shown the existence of different types of traders, or market actors, that have stable relationships with fishers, sometimes referred to as "patrons"; such traders often buy and/or sell fish to each other, and do so through stable trading arrangements. Our study increases understanding of the influence of such regional or horizontal relationships between traders (often named patrons or middleman), and shows that they have potential implications for both the individual- and system-level dynamics of fish provision. In addition, we highlight how traders can have different roles depending on their connectivity patterns. Understanding the different roles of traders will be key to better account for traders and post-harvesting activities in fisheries decision-making processes [15,22,78].

To our knowledge, the Small-Trade model presents one of the first efforts that contributes to an understanding of how diversification strategies (in species and spatial trade relationships) interact in trade networks and influence fish provision in the spatially-heterogeneous, multi-species, and multi-market context that often characterizes small-scale fisheries. Through a multi-method modelling approach that pushes the frontiers of integrating agent-based modeling and network analysis, this study contributes to an emerging research agenda to investigate fish trade networks in small-scale fisheries. We provide insights on processes that characterize trade in an SSF context, and how such processes can lead to spillover effects between species and regions. Our model provides key insights to inform future empirical research aimed at understanding the effects of regional trade in fish provision, and acknowledging the diversity of trader types and their horizontal relationships. It provides key considerations that evidence the need to better understand spatial heterogeneity and connectivity in regional trade networks, the influences of different types of social and environmental change on fish provision, and traders' agency in relation to their position in trade networks embedded in social relationships.

## Supporting information

**S1 Appendix. ODD+D protocol.**
(DOCX)

**S2 Appendix. Empirical analysis of interview data.**
(DOCX)

**S3 Appendix. Network analysis and generation of experiments for the ABM.**
(DOCX)

**S4 Appendix. Sensitivity analysis.**
(DOCX)

## Acknowledgments

We thank all interviewed participants that contributed to this research. We thank Juan Rocha and Sofia Käll for their reviews of this manuscript, Sonja Radosavljevic for mathematical explorations around the decision-making model, and Kara Pellowe for proofreading the manuscript.

## Author Contributions

**Conceptualization:** Blanca González-Mon, Emilie Lindkvist, Örjan Bodin, Maja Schlüter.

**Data curation:** Blanca González-Mon.

**Formal analysis:** Blanca González-Mon.

**Funding acquisition:** Maja Schlüter.

**Investigation:** Blanca González-Mon, Emilie Lindkvist, José Alberto Zepeda-Domínguez, Maja Schlüter.

**Methodology:** Blanca González-Mon, Emilie Lindkvist, Örjan Bodin, Maja Schlüter.

**Project administration:** Blanca González-Mon.

**Software:** Blanca González-Mon, Emilie Lindkvist.

**Supervision:** Emilie Lindkvist, Örjan Bodin, José Alberto Zepeda-Domínguez, Maja Schlüter.

**Validation:** Emilie Lindkvist, Örjan Bodin, Maja Schlüter.

**Visualization:** Blanca González-Mon.

**Writing – original draft:** Blanca González-Mon.

**Writing – review & editing:** Emilie Lindkvist, Örjan Bodin, José Alberto Zepeda-Domínguez, Maja Schlüter.

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
