## [Decision Letter · Decision Letter 0]

14 May 2021

PONE-D-21-08237

Fish provision in a changing environment: the buffering effect of regional trade networks

PLOS ONE

Dear Dr. Gonzalez-Mon,

Thank you for submitting your manuscript to PLOS ONE. After careful consideration, we feel that it has merit but does not fully meet PLOS ONE’s publication criteria as it currently stands. Therefore, we invite you to submit a revised version of the manuscript that addresses the points raised during the review process.

I'd like to note that both reviewers were quite positive about the quality of the study, but had similar comments regarding the need to improve the clarity. Carefully addressing the comments/suggestions of Reviewer 1 in particular should help you to focus and improve the manuscript and hopefully increase its impact. Moving some key information from the supporting information to the main text may also help.

We look forward to receiving your revised manuscript.

Kind regards,

Carla A Ng

Academic Editor

PLOS ONE

Journal Requirements:

3. Please upload a copy of Supporting Material 2 which you refer to in your text on page 37.

Reviewers' comments:

Reviewer's Responses to Questions

**Comments to the Author**

1. Is the manuscript technically sound, and do the data support the conclusions?

Reviewer #1: Yes

Reviewer #2: Yes

2. Has the statistical analysis been performed appropriately and rigorously? 

Reviewer #1: N/A

Reviewer #2: Yes

3. Have the authors made all data underlying the findings in their manuscript fully available?

Reviewer #1: Yes

Reviewer #2: Yes

4. Is the manuscript presented in an intelligible fashion and written in standard English?

Reviewer #1: Yes

Reviewer #2: Yes

5. Review Comments to the Author

Reviewer #1: This is an interesting paper exploring the effects of regional trade on outcomes of a stylized small-scale fishery, with trade network aspects and some of the trader behaviors based on empirical data and interviews in Baja California Sur, Mexico. This is a worthwhile topic, and the authors have undertaken a rich mixed-methods approach.

The analysis seems technically sound, and therefore I do not have concerns that should preclude publication in PLoS One. However, I offer the following comments, which the authors may take or leave in a revision, as they see fit.

Big-picture comment:

My main critique of the paper is that it would benefit from more focus and clarity of scope, at least in how it is written (if not the study itself). In some ways, the model is tailored to a specific site; in other ways it is general and highly abstract (e.g., having two regions and two species). In which light should readers be interpreting it? There are a huge number of results presented, but it was hard to discern a clear punchline or take-home message. The intuition of key results was also sometimes hard to discern. The ODD+D protocol is very helpful, but the main text does not describe the key assumptions especially clearly. E.g., it was not made clear until quite late in the paper how fishing mortality is determined. Some of the quantitative network analyses seemed unnecessary and tangential.

I would recommend that the authors attempt to address this in a revision, as it would increase the impact of the paper. But I also think that, by what I understand are PLoS One’s editorial guidelines (i.e. papers must be technically correct with no additional filters), the paper is acceptable in its current form.

Minor comments:

Line 140: I suggest explaining here that the fishers are not represented (as currently done on line 177). I found myself wondering who was fishing while reading these lines in between, which was distracting. I also suggest previewing in these paragraphs how the fishing pressure dynamics are governed, since the Abstract mentions an effect on overexploitation, which implies a feedback between agent decisions and fishing pressure.

Line 223: It’s ok to put some of these details in the SI, but the authors should put some basic information about agent decision making (e.g. what is their assumed objective?) in the main text, since the agent decision-making priorities are central drivers of any results.

Line 280: What is being varied among the 300 runs? Starting conditions? Parameters? Please clarify.

Line 461: What is the intuition of this result? The idea that the trade network increases efficiency for the seasonal species is intuitive. This second result, less so, especially as it relates to waste. Why would trade ever increase waste? I’m sure there’s a reason that makes sense, but it would be helpful to explain it.

Acknowledgements: It looks like there are placeholders (“NAME”) left in here inadvertently.

What are the red error bars in Fig. 4 showing? The caption does not make this clear.

Reviewer #2: This study is about the development and application of an agent-based model which represents fish trade networks in small-scale fisheries, and how they interact with one another. The study and model are somewhat abstract but based on empirical data from a fishery in Baja California Sur. Overall, the study, modeling approach, assumptions, data analysis, findings, and writing all reflect quality work, and I recommend this manuscript for publication.

A few general comments:

The manuscript and the model refer to fish populations – how are these populations modeled? I assumed that detailed biology is not included, since the focus and thus detail of the model is on the trading behaviors, however it is not very clear to the reader where the fish that are traded come from. Using the terms fish populations, seem to allude to a biological sub-model that is part of the simulation but not well described in the paper. I do note that some descripting is buried in Table 2, where it states that the fish stock regenerates according to a logistic growth function. This information however should be part of the body of the text somewhere, and the parameters of the logistic function should be included. The rate of growth would directly influence the fish available to the trade networks and thus have downstream impacts on your results.

To fully understand this study, one needs to also read the González-Mon et al. (2016) manuscript. Is there a way to include more information about the survey done in that study which you are using? For example, what were the questions asked (could be added to the supplementary material), what was the sample size, what were the general findings, and how is the data from the González-Mon et al. study being using in this study? I hate to suggest this given that the manuscript and supplemental material are already quite long, however it would be useful for the reader.

Line specific comments:

Lines 114 through 122 can be deleted given that the sections are sufficiently defined in the manuscript

Line 225 – sentence is incomplete

Line 278 – please define the environmental dynamics you are discussing or refer the readers to table 4 in the supplementary material section 5

Line 301 – double check the formula – I don’t think you need the minus one

Lines 332 through 333 – delete, not needed

Line 379 – can you provide the formula for the decision-making function?

Line 579 – change to “…emphasize that not only may fishers diversify…” [word “may” before “fishers”]

Line 607 – delete the word “can”

6. PLOS authors have the option to publish the peer review history of their article (what does this mean?). If published, this will include your full peer review and any attached files.

Reviewer #1: No

Reviewer #2: No

---

## [Author Response · Author response to Decision Letter 0]

18 Aug 2021

Our response is added directly after each point raised by the academic editor and reviewers.

Academic editor: 

I'd like to note that both reviewers were quite positive about the quality of the study, but had similar comments regarding the need to improve the clarity. Carefully addressing the comments/suggestions of Reviewer 1 in particular should help you to focus and improve the manuscript and hopefully increase its impact. Moving some key information from the supporting information to the main text may also help.

Response: Dear Editor, thank you very much for your comments and for considering our manuscript for publication. We have now addressed the reviewers´ comments one by one as you may see below, and paid special attention to the comments highlighted by reviewer 1 to increase the clarity and potential impact of the manuscript.

Reviewer #1: 

This is an interesting paper exploring the effects of regional trade on outctomes of a stylized small-scale fishery, with trade network aspects and some of the trader behaviors based on empirical data and interviews in Baja California Sur, Mexico. This is a worthwhile topic, and the authors have undertaken a rich mixed-methods approach.

The analysis seems technically sound, and therefore I do not have concerns that should preclude publication in PLoS One. However, I offer the following comments, which the authors may take or leave in a revision, as they see fit.

Response: Thank you very much for your comments and recommending this article for publication. Your comments have been very useful in improving the manuscript, and we respond to each and one of them bellow. Note that the line numbers indicated below refer to the reviewed manuscript with track changes. We hope these changes address the concerns of the reviewer.

Big-picture comment:

My main critique of the paper is that it would benefit from more focus and clarity of scope, at least in how it is written (if not the study itself). In some ways, the model is tailored to a specific site; in other ways it is general and highly abstract (e.g., having two regions and two species). In which light should readers be interpreting it? There are a huge number of results presented, but it was hard to discern a clear punchline or take-home message. The intuition of key results was also sometimes hard to discern. The ODD+D protocol is very helpful, but the main text does not describe the key assumptions especially clearly. E.g., it was not made clear until quite late in the paper how fishing mortality is determined. Some of the quantitative network analyses seemed unnecessary and tangential.

Response: Thank you for your useful comments, we have addressed this feedback in several ways: 1) we have added a paragraph explaining our approach and guiding the reader to how it should be interpreted according to the deliberate mix between empirical and theoretical components (lines 140-154); 2) we have reviewed the results section (4 and 5) and removed any result that seemed tangential, clarified the explanations where needed for those results that were potentially not intuitive (e.g. lines 915-928), added a new subheading (section 5.2) to highlight more clearly the three key take-home messages of section 5, and clarified the language throughout; 3) we have tried to clarify the assumptions in the main text wherever possible, please see some of these clarifications as a response to your other comments below. If there is something that would still require further clarification, please do not hesitate to let us know. We have also reviewed the supplementary material 3 to remove any results from the network analysis that were unnecessary for this manuscript, but we have mostly kept the key results for transparency following PLoS One guidelines.

I would recommend that the authors attempt to address this in a revision, as it would increase the impact of the paper. But I also think that, by what I understand are PLoS One’s editorial guidelines (i.e. papers must be technically correct with no additional filters), the paper is acceptable in its current form.

Response: Thank you very much for your positive comment and for your recommendation for publication. We have now worked in the manuscript to try to clarify the message and increase the readability and, hopefully, its impact.

Minor comments:

Line 140: I suggest explaining here that the fishers are not represented (as currently done on line 177). I found myself wondering who was fishing while reading these lines in between, which was distracting. I also suggest previewing in these paragraphs how the fishing pressure dynamics are governed, since the Abstract mentions an effect on overexploitation, which implies a feedback between agent decisions and fishing pressure.

Response: Thank you very much for this suggestion. We have moved up the clarification about the fishers (lines 199-202) and clarified the assumptions about the fishing pressure in the text in that section. For instance, we made explicit the formula on the fishing pressure dynamics in line 235 and clarified the origin and meaning of the parameter values of growth rate and carrying capacity in the model in that same section. 

Line 223: It’s ok to put some of these details in the SI, but the authors should put some basic information about agent decision making (e.g. what is their assumed objective?) in the main text, since the agent decision-making priorities are central drivers of any results.

Response: Thank you for pointing this out. We have now clarified the agent’s objectives from early on (lines 356-357). However, we would also like to note that one of the research questions we put forward is to identify the trading processes that govern the trade interactions, and we present the answers as a result in section 4. Therefore much of the descriptions and explanations to understand the trading dynamics in the model are included as a result in section 4 deliberately, since including them in the method description (section 3) would undermine the multi-method approach that we followed where those dynamics were decided as a result of the qualitative data analysis. We have now clarified in the previously numbered line 223 that these dynamics are presented in section 4 (see lines 430-431).

Line 280: What is being varied among the 300 runs? Starting conditions? Parameters? Please clarify.

Response: Thank you very much for pointing this out, we have now clarified this in lines 512-514. We perform 300 runs per scenario and network type to account for stochasticity in the model, which is a common approach when running agent-based models. The model settings (initial conditions, parameters) of each scenario and network type are exactly the same in the 300 runs but the randomness/stochasticity can still generate different outcomes. This is why the panel figures generally represent averages across the 300 runs, but for figure 5 where we analyze the effect of the random allocation of traders in different regions as explained in the figure caption. 

Line 461: What is the intuition of this result? The idea that the trade network increases efficiency for the seasonal species is intuitive. This second result, less so, especially as it relates to waste. Why would trade ever increase waste? I’m sure there’s a reason that makes sense, but it would be helpful to explain it.

Response: Thank you for pointing this out. These dynamics result because of the dynamics between the two species and the need to allocate a limited effort between the two. Trade makes the market of one species more efficient, but less efficient (more scarcity, more waste) for the other species. In order to clarify the message and remove tangential results, we have decided to take away the explanations of the fish waste dynamics from section 5.1. where this sentence belonged, since these dynamics are already reflected by the scarcity results and that sentence was indeed confusing.

Acknowledgements: It looks like there are placeholders (“NAME”) left in here inadvertently.

Response: Thank you, this has been addressed now.

What are the red error bars in Fig. 4 showing? The caption does not make this clear.

Response: Thank you for pointing this out, we have clarified this in the figure caption now. The red error bars or intervals represent the variability (standard deviation) in scarcity over the last 5 years of the simulation and across the 300 runs, since the grey bars represent the average values.

Reviewer #2: 

This study is about the development and application of an agent-based model which represents fish trade networks in small-scale fisheries, and how they interact with one another. The study and model are somewhat abstract but based on empirical data from a fishery in Baja California Sur. Overall, the study, modeling approach, assumptions, data analysis, findings, and writing all reflect quality work, and I recommend this manuscript for publication.

Response: Thank you very much for your positive comment and for all of your suggestions below, they are very much appreciated. We have replied to each of the comments below, and hope that our responses address your concerns. Note that the line numbers indicated below refer to the reviewed manuscript with track changes.

A few general comments:

The manuscript and the model refer to fish populations – how are these populations modeled? I assumed that detailed biology is not included, since the focus and thus detail of the model is on the trading behaviors, however it is not very clear to the reader where the fish that are traded come from. Using the terms fish populations, seem to allude to a biological sub-model that is part of the simulation but not well described in the paper. I do note that some descripting is buried in Table 2, where it states that the fish stock regenerates according to a logistic growth function. This information however should be part of the body of the text somewhere, and the parameters of the logistic function should be included. The rate of growth would directly influence the fish available to the trade networks and thus have downstream impacts on your results.

Response: Thank you very much for your suggestion. There are four simple biological sub-models (with the same formula and values) representing the fish stocks and we have now clarified and added that formula of the fish population dynamics in section 2 (line 235). However, a more detailed biological model is not included, and therefore we now refer to the fish populations as fish stocks throughout the manuscript to avoid confusion. We have also clarified the origin and meaning of the parameter values of growth rate and carrying capacity in the model, and explicitly explained how the model was calibrated so that parameters are relative to each other. Because the market demand and the harvesting effort are calibrated to keep the stock at a level that provides the maximum sustainable yield over time during baseline model conditions, changes in the growth rate parameter will be accounted for by the way we set the harvesting levels and do not impact the qualitative dynamics described in the results. This calibration and parameter choice allowed us to focus on the trading structures and dynamics since this model does not aim to explain the influence of different biological parameters. We have explained this in more detail in the main manuscript (lines 237-247). We are sincerely sorry for the confusion our vagueness on this point created for the reviewer.

To fully understand this study, one needs to also read the González-Mon et al. (2016) manuscript. Is there a way to include more information about the survey done in that study which you are using? For example, what were the questions asked (could be added to the supplementary material), what was the sample size, what were the general findings, and how is the data from the González-Mon et al. study being using in this study? I hate to suggest this given that the manuscript and supplemental material are already quite long, however it would be useful for the reader.

Response: Thank you very much for this suggestion. We have added more details regarding the data collection and analysis used in the previous paper in the supplementary material 3 to address this request. Due to the length of the manuscript and to maintain the readability and clarify of findings specific to this study, we have added most of the information in the supplementary material instead of in the main text, but we have also added a few clarifications in the main text were relevant (e.g. line 450-452) and references to the supplementary material were appropriate. We believe the information currently added in the manuscript should provide enough detail to understand this study and not require to read González-Mon et al. (2019), but we will be happy to add more details if needed.

Line specific comments:

Lines 114 through 122 can be deleted given that the sections are sufficiently defined in the manuscript

Response: Thank you, this has been addressed now.

Line 225 – sentence is incomplete

Response: Thank you, this has been clarified now.

Line 278 – please define the environmental dynamics you are discussing or refer the readers to table 4 in the supplementary material section 5

Response: Thank you for pointing this out, we refer now to table 4 in the supplementary 1 for details, and we have also added a more specific definition in the main text and in table 1 of the main text (see lines 494-510).

Line 301 – double check the formula – I don’t think you need the minus one

Response: Thank you for pointing this out, the formula was verified and it is correct. The -1 here was included to center the value around 0 and make it more intuitive, so that MSY is 0%, and under or over exploitation can be referred to that baseline. We have clarified this now in the manuscript (lines 548-549). We are sorry for the unclarity.

Lines 332 through 333 – delete, not needed

Response: Thank you, this has been deleted now.

Line 379 – can you provide the formula for the decision-making function?

Response: We agree with the reviewers that it would be appropriate to add a simple formula for the decision-making model. However, it is not possible to provide a simple formula as it is a quite complex algorithm in that it consists of a decision tree that are better represented as sequences of if-then statements. As such, we present it through pseudo code and now name it algorithm in the manuscript to clarify the implementation of this decision-making model. We have added the part of the pseudo code for the decision-making algorithm in the manuscript now (lines 710-758), and also added the simplified pseudo code with a longer explanation to the supplementary material (and indicated so in line 711) as follows: 

If 〖de〗_A>〖pe〗_A then e_A=〖de〗_A-〖mag〗_A

If 〖de〗_A<〖pe〗_A then e_A=〖de〗_A+〖mag〗_A

If 〖de〗_A=〖pe〗_(A) then e_A=〖de〗_A

where 

〖mag〗_A=br×abs (〖de〗_A-〖pe〗_A) 

Where deA denotes the optimal effort based on meeting the demand for species A according to the demand-driven sub-model; peA, the optimal effort based on maximizing the CPUE for species A according to the CPUE-driven sub-model. The influence, or weight, of each sub-model is determined by a balance rate (br, with range from 0 to 1). magA represents the magnitude of the effort difference between the demand versus the CPUE optimal efforts. eA denotes the new balanced effort for species A. This decision-making algorithm is repeated for species B, where eB denotes the new balanced effort for species B. The final effort is then balanced between the species in relation to E:

e_(A_new)=E/((e_A+e_B)) e_A 

e_(B_new)=E/((e_A+e_B)) e_B

where eA_new and eB_new are the actual efforts applied for species A and B respectively, and E is the maximum effort that the trader can apply and need to share between species A and B.

Please refer to SM5 in section 7 of S1 Appendix for further details. If you consider that this adds too much complexity to the manuscript and hinders readability we would also be willing to remove it from the main manuscript and explain it in S1 Appendix. We would also be pleased to address any further questions you may have about the decision-making algorithm.

Line 579 – change to “…emphasize that not only may fishers diversify…” [word “may” before “fishers”]

Response: Thank you, this has been changed now.

Line 607 – delete the word “can”

Response: Thank you, this has been changed now.

---

## [Editor Report · Decision Letter 1]

6 Dec 2021

Fish provision in a changing environment: the buffering effect of regional trade networks

PONE-D-21-08237R1

Dear Dr. Gonzalez-Mon,

We’re pleased to inform you that your manuscript has been judged scientifically suitable for publication and will be formally accepted for publication once it meets all outstanding technical requirements.

Kind regards,

Carla A Ng

Academic Editor

PLOS ONE

Additional Editor Comments (optional):

Thank you for submitting your point-by-point responses and revisions, as well as for your patience with the review process. It had been my hope to have the original reviewers re-review the manuscript, but as neither were available I performed the review myself. My opinion is that you have adequately addressed all the reviewer's comments and that your revisions have aided in improving the clarity of the manuscript. I am therefore Accepting the manuscript for publication.
---

## [Editor Report · Acceptance letter]

9 Dec 2021

PONE-D-21-08237R1 

Fish provision in a changing environment: the buffering effect of regional trade networks 

Dear Dr. Gonzalez-Mon:

I'm pleased to inform you that your manuscript has been deemed suitable for publication in PLOS ONE. Congratulations! Your manuscript is now with our production department. 

Kind regards, 

on behalf of

Dr. Carla A Ng 

Academic Editor

PLOS ONE